# A Study on the Impact of Watershed Compensation Policies on Green Technology Innovation Ecosystems

**Mo Li [1], Jianhua Zhu [2],* and Hua Dong [1]**

1. School of Economics and Management, Qingdao University of Science and Technology, Qingdao 266061, China; 0019070001@mails.qust.edu.cn (M.L.); donghua@qust.edu.cn (H.D.)
2. School of Economics and Management, Harbin Institute of Technology, Weihai 264209, China
* Correspondence: zhujh@hit.edu.cn

**Abstract:** This study uses the implementation of the watershed compensation policy as a quasi-natural experiment and selects a sample of 53 cities located within the four major watersheds from 2005 to 2022. By employing a staggered difference-in-differences model and a synthetic control difference-in-differences model, the study investigates how watershed compensation policies influence green technology innovation ecosystems and delves into the underlying mechanisms that are responsible for these impacts. The research reveals the following findings: (1) The introduction of the watershed compensation policy markedly boosts the development of green technology innovation ecosystems in the pilot cities, and this finding remains consistent following a series of robustness checks. (2) An analysis of the mechanisms indicates that the watershed compensation policy exerts its impact on the advancement of green technology innovation ecosystems through a reduction in carbon emission intensity and the enhancement of wastewater treatment efficiency. (3) The influence of the watershed compensation policy on green technology innovation ecosystems varies according to the level of public financial expenditure and labor productivity. This research offers a factual foundation for comprehending the effects of watershed compensation policies on the innovation of green technologies within China.

**Keywords:** watershed compensation policy; green technology innovation; staggered double-difference modeling; synthetic double-difference modeling

## 1. Introduction

As a special natural geographic unit (Ren, 2020 [1]), a watershed is an ecosystem centered around water resources as its primary medium (Hu Zhenhua, 2016 [2]), supporting various human economic and social activities, and water is intricately linked to the economic and social progress of the watershed, as well as to human activities and livelihoods (Ali R, 2019 [3]). As the watershed experiences rapid economic and social development alongside significant population growth, the intensity of human exploitation of water resources and the resulting damage to the water environment have steadily increased, making water pollution in the watershed an increasingly pressing issue. At the end of the 1990s, the mechanism of ecological compensation was introduced into the field of watershed ecological and environmental management (Gao, 2019 [4]). Watershed ecological compensation plays a vital role in balancing regional interests, addressing water pollution issues within the watershed, and promoting the sustainable economic and social development of a region (Ren, 2020 [1]). Enacting watershed ecological compensation policies has altered the customary ways of living and working among local communities,



leading to notable advancements in curbing water pollution, elevating the water purity, and augmenting the value of ecosystem services in the watershed (Ren, 2020 [5]).

The ecological compensation of the Xin'an River Basin is part of the pioneering inter-provincial ecological compensation program in China (Li Tan, 2022 [6]). In 2011, a pioneering horizontal ecological compensation pilot agreement for the basin was established through a signing between Anhui and Zhejiang provinces. The policy mainly aims to improve the water environment by managing urban and rural domestic sewage, controlling industrial point source pollution, and addressing rural non-point source pollution (Yi, 2022 [7]). The pilot advocates market-based horizontal compensation, supplemented by vertical fiscal transfers (Wang Xiaoli, 2018 [8]), and through the comprehensive application of various environmental economic instruments (CLOT, 2017 [9]), the standard for calculation is based on the water quality at the cross-sections where the river basin provinces meet. The downstream area will give the upstream area CNY 100 million in compensation if the upstream provides a better-than-basic standard, and the downstream area will give the upstream CNY 100 million in compensation if the upstream provides a worse-than-basic standard; the upstream area will give the downstream area CNY 100 million in compensation if the upstream area provides a worse-than-basic standard; and the upstream area will give the downstream area CNY 100 million in compensation if the upstream area provides a worse-than-basic standard. If the upstream water quality falls below the basic standard, the downstream area will provide CNY 100 million to the upstream area. However, if the water quality meets the basic standard, no compensation will be exchanged between the upstream and downstream areas (Jin, 2022 [10]). This "betting" agreement motivates participants to diligently protect the basin's ecosystem, promote the sustainable exploitation of water resources, and maintain the ongoing delivery of aquatic ecological services (Jing, 2018 [11]). This approach ultimately aims to foster harmonious development among the economy, society, and environment. As a result, a principal goal of the watershed ecological compensation policy is to boost the efficiency of water resource use while upholding the principles of sustainable development.

Thus, the official initiation of watershed eco-compensation practices in China has been realized, and since then, the Jiuzhou River, the Tingjiang-Hanjiang River, and the Dongjiang River watersheds have been approved as pilot watershed eco-compensation basins in turn, gradually exploring a sustainable and high-quality development path. As an integrated initiative, the watershed ecological compensation mechanism prioritizes environmental protection as its foundation, emphasizes ecological compensation as a core component, and adopts green development as its guiding approach. It leverages market regulation and resource allocation to drive industrial transformation and upgrading, fostering high-quality economic development (Wang, 2022 [12]). A critical question is whether the region receiving compensation can improve the ecological environment of the watershed and, concurrently, optimize its regional industrial structure. Investigating the effects of watershed ecological compensation policies on industrial restructuring in the compensated areas is of considerable theoretical and practical importance. Such an examination allows for an accurate and thorough evaluation of the economic advantages of implementing watershed ecological compensation, which in turn assists in developing a solid and efficient policy framework.

Nations that are experiencing economic transition prioritize innovation as a key to harmonizing ecological and economic interests, a crucial and evident role. Addressing the pronounced imbalance between these, establishing a green tech innovation ecosystem is essential for advancing sustainable economic growth. This ecosystem, under strategic innovation guidance, exhibits enhanced coordination, self-regulation, and openness, becoming a central force in industry reorganization and economic shifts. Amid economic

challenges, understanding the impact of watershed compensation policies on green innovation is vital. These policies, as economic incentives, balance regional ecological and economic interests, spurring corporate green tech innovation. Amidst the escalating climate and environmental crises, balancing watershed conservation with economic greening is an urgent theoretical and practical challenge. The synergy between watershed policies and green innovation ecosystems offers a fresh approach to this issue, showcasing the potential for policy and tech integration. This alliance fortifies watershed conservation and supports the growth of green innovation ecosystems, laying a foundation for economic transformation and sustainability.

Based on the pilot policy of "watershed compensation", a quasi-natural experiment was established. Compared with the existing literature, the main contributions of this article are as follows: firstly, through the mechanism analysis of the impact of the watershed compensation policy on the green technology innovation ecosystem, it was found that the watershed compensation policy improves the construction level of the green technology innovation ecosystem in the pilot area by reducing carbon emissions and improving the sewage treatment efficiency; secondly, this article explores the significant heterogeneity in the impact of watershed compensation policies on green innovation systems at the levels of public expenditure and labor productivity; thirdly, the article demonstrates a certain degree of innovation in the selection of explanatory and mediating variables. And the innovative combination of the synthetic control method and interleaved dual activation model validates the robustness of the research content of this article.

## 2. Theoretical Analysis and Hypothesis Development

### 2.1. Influence of Watershed Compensation Policies on Green Innovation Ecosystems in Compensated Regions

Companies serve as pivotal catalysts for regional technological advancements and green innovation solutions. Research by Shen Neng (2012) [13] and others indicates that enterprises' profit and production functions are intricately linked to their emission levels and environmental protection expenditures. Specifically, pollutant emissions positively correlate with output revenue and negatively with environmental protection expenditures, suggesting that increased investments in environmental protection leads to reduced emissions. The implementation of eco-compensation initiatives, particularly watershed compensation policies, significantly impacts green innovation systems. These policies not only incentivize enterprises to invest in pollution control, mitigating environmental degradation and fostering the "technological progress effect of pollution control", but also encourage technological innovation through financial incentives. The "innovation compensation effect" arises as upstream enterprises are rewarded for adopting environmentally friendly practices that benefit downstream areas, motivating them to innovate not solely for compliance but also to capitalize on government incentives.

Moreover, the integration of watershed compensation policies into green innovation systems promotes collaboration among upstream and downstream enterprises, as well as between enterprises and government agencies. This collaborative framework facilitates the sharing of best practices, the development of novel technologies, and the optimization of resource use, collectively contributing to a more sustainable and innovative production process. Based on these premises, the inaugural hypothesis of this study is put forth:

**H1:** *The policy of watershed compensation is posited to markedly boost the caliber of green technology innovation within compensated regions, thereby fortifying the evolution of the innovation ecosystem.*

*2.2. Mechanisms of Impact of Watershed Compensation Policies on Innovation Ecosystems in Compensated Areas*

Under the cross-provincial horizontal ecological protection compensation mechanism, local governments re-evaluate the equilibrium between economic growth and ecological preservation, thereby strengthening environmental regulations and implementing measures for both end-of-pipe treatment and source prevention to reduce water pollutant emissions, in accordance with agreed-upon watershed water quality standards. Notably, adopting a source prevention and control approach for water pollution treatment can synergistically drive $CO_2$ reductions. Within the framework of horizontal inter-provincial ecological protection compensation systems for river basins, local governments vigorously promote cleaner production (Wang, 2018 [8]). This, in turn, encourages enterprises to utilize cleaner energy, enhance their energy conservation, and augment investments in green innovations. Consequently, the industry's overall energy structure improves, the energy intensity decreases, and $CO_2$ emissions are reduced. These actions not only mitigate the environmental impact but also bolster the construction of a green ecosystem. Illustratively, the Xin'an River Basin has optimized and upgraded over 510 projects during the first two rounds of the agreement, investing CNY 6 billion in constructing a circular economy park with integrated heating, desalination, and sewage treatment systems. This has facilitated the establishment of a green industry system, resulting in significant economic growth—with the GDP surpassing the CNY 50-billion and 60-billion milestones—and the financial income exceeding CNY 200 yuan. This exemplifies the qualitative and effective transformation of green resources into economic benefits, achieving substantial improvements in ecological, economic, and social outcomes.

Similarly, the Jiuzhou River Basin has upgraded nine water-related industrial enterprises as of 2022, implemented "one factory, one case" pollution control for 16 water-related enterprises, and constructed a 73.33-hectare small and medium-sized enterprise industrial transfer park in the Upper Jiuzhou River Basin. By guiding water-related enterprises to relocate from the riverbanks into the park and supporting the development of green environmental protection industries, the basin has promoted industrial transformation and upgrading. These initiatives have contributed to reducing carbon emissions, thereby enhancing the resilience and sustainability of the green ecosystem (Chen, 2022 [14]).

Conversely, within the framework of ecological compensation for inter-provincial river basins on a horizontal level, local governments can restrict the development of enterprises in water-pollution-intensive industries, which are often highly carbon-intensive as well (Wang, 2018 [8]). This regulation not only controls the market entry of related enterprises but also promotes the development of the entire industry's energy structure and intensity towards $CO_2$ reductions. Additionally, enhancing wastewater treatment rates through compensation mechanisms further refines green ecosystem construction. For instance, the Xin'an River Basin enforces a negative industrial entry list system and invests heavily in pollution prevention, resulting in the closure and relocation of numerous polluting enterprises. Similarly, the Jiuzhou River Basin shuts down polluting industries and refuses to approve wastewater-discharging projects (Tu 2012 [15]). Moreover, the Hanjiang-Tingjiang River Basin and Longyan City in Fujian Province conduct extensive remediation projects, which include improving wastewater treatment facilities to enhance treatment rates. By integrating wastewater treatment improvements into ecological compensation strategies, these regions effectively bolster their green ecosystems, ensuring sustainable development and environmental protection. The second set of hypotheses of this paper can be proposed as follows:

**H2a:** *Watershed offset policies can improve local innovation ecosystems by reducing carbon emissions in compensated areas.*

**H2b:** *Watershed compensation policies can improve local innovation ecosystems by increasing wastewater treatment rates in compensated areas.*

*2.3. The Diversity in the Influence of Watershed Compensation Policies on the Development of Green Technology Innovation Ecosystems*

In regions with high public fiscal expenditures and high labor productivity, the effectiveness of river basin compensation in enhancing green ecosystem construction is more pronounced. High fiscal revenues enable governments to allocate more funds towards long-term development projects (Yan, 2024 [16]), such as education and scientific research, which directly support green technology R&D and innovation. This fosters a conducive environment for technological advancements, providing sustained funding and intellectual resources. Furthermore, high labor productivity not only augments firm profits but also encourages investments in technological innovation (Brynjolfsson, 2011 [17]), as firms recognize the correlation between productivity gains and enhanced competitiveness. With a skilled and efficient workforce, enterprises possess the necessary human capital for technological breakthroughs, further supporting green ecosystem development (Fedulova, 2019 [18]). Thus, the combination of robust public fiscal expenditures and high labor productivity synergistically amplifies the impact of river basin compensation on green ecosystem construction. The advantages of resource-based cities in terms of the impact of watershed compensation policies on green technology innovation ecosystems are mainly reflected in the fact that they can obtain financial support through compensation policies for green technology innovation; as a provider and protector of ecological services, there is a stronger willingness to improve the resource utilization efficiency and achieve sustainable development through technological innovation. Based on this, the following hypotheses are proposed:

**H3a:** *Cities with greater public financial outlays experience a more pronounced impact of watershed compensation policies on their green technology innovation ecosystems.*

**H3b:** *In cities where the labor productivity is higher, the impact of watershed compensation policies on green technology innovation ecosystems is more pronounced.*

**H3c:** *In resource-based cities, watershed compensation policies have a more significant impact on the green technology innovation ecosystem.*

Following the above ideas, this article constructs a mechanism framework for the impact of watershed compensation policies on the green technology innovation ecosystem, as shown in Figure 1.

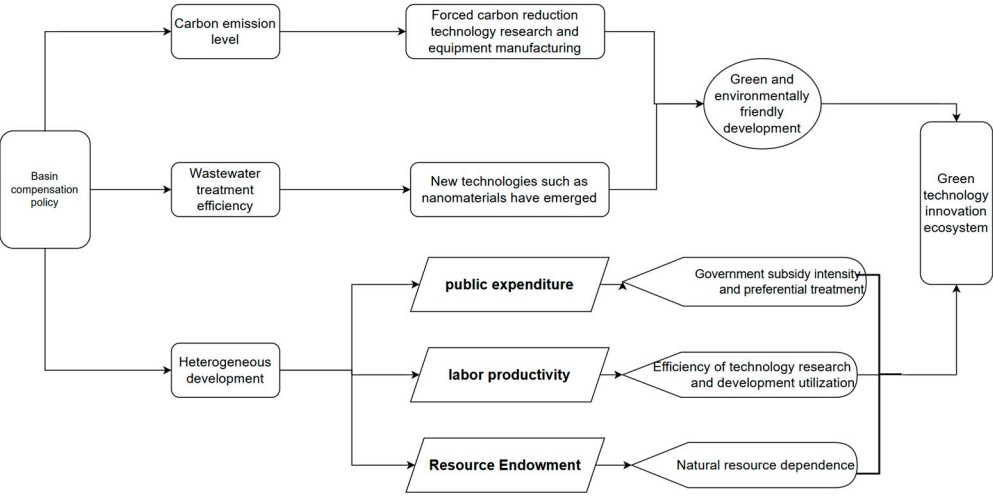

**Figure 1.** Mechanism diagram.

## 3. Research Methodology and Data Sources

### 3.1. Model Setup

#### 3.1.1. Interleaved Double-Difference Models

DID analysis with staggered implementation is suitable for situations where members of the treatment group are subjected to the intervention at various times. Since there are time batch differences in the implementation of watershed compensation in the major cities of each watershed, to investigate the effects of this policy on the green technology innovation ecosystem, this paper refers to the staggered double-difference method proposed by Beck (2010) [19] and others and adopts the two-way fixed corresponding model, using the pilot policy of the national-level big data comprehensive experimental zone as a quasi-natural experiment; the impact of this policy on provincial climate change is then analyzed. The empirical model is presented in the following equation:

$$Y_{it} = \alpha_0 + \alpha_1 \text{policy}_i \times \text{post}_t + \alpha_2 X_{it} + \mu_i + \delta_t + \varepsilon_{it}$$

In the equation, i and t denote the province and year, respectively; the explanatory variable $Y_{it}$ denotes the climate level of province i in year t; $\text{policy}_i$ is the policy dummy variable, which takes the value of 1 when a city is a compensated city of the watershed compensation policy and 0 otherwise; $\text{post}_t$ is the time of policy implementation dummy variable, which takes the value of 1 in the year of the implementation of the watershed compensation policy and thereafter and 0 otherwise; $\text{policy}_i \times \text{post}_t$ is the interaction term of the two dummy variables; the coefficient $\alpha_1$ reflects the effect of the implementation of the watershed compensation policy on the green technology innovation ecosystem; $\alpha_0$ and $\alpha_2$ denote the constant term and the parameter estimation of the control variables, respectively; H is the control variable of the green technology innovation ecosystem; $\mu_i$ and $\delta_t$ represent the individual fixed effect and the time-fixed effect, respectively; and $\varepsilon_{it}$ stands for the random interference term.

#### 3.1.2. Synthetic Control Double-Difference

The synthetic control method (SDID) is a treatment effect model that is designed for panel data. It calculates the treatment effect by examining a policy's impact through a comparison of the double-differences between the treatment group and a synthetic control group, both pre- and post-intervention (Arkhangelsky, 2021 [20]). The synthetic control group is constructed by selecting individuals who did not adopt the policy and applying an optimal weighting algorithm based on the characteristics of the treatment group prior

to policy implementation. The model for the synthetic control double-difference method SDID is as follows:

Consider a panel in equilibrium: n individuals, T periods, with individual i's outcome variable in period t being $Y_{it}$ and the binary treatment variable being $W_{it} \in \{0, 1\}$. Also assume that the first $N_{co}$ individuals are not treated. The latter $N_{tr} = N - N_{co}$ individuals will be treated at $T_{\text{pre}}$, which is similar to a synthetic control SC.

First, we look for weights $\hat{\omega}_i^{\text{sdid}}$ so that the ex ante trend of the yes-treatment group and the ex ante trend of the control group are as similar as possible, i.e., $\sum_{i=1}^{N_\infty} \hat{\omega}_i^{\text{sdid}} Y_{it} \approx N_{tr}^{-1} \sum_{i=N_\infty+1}^{N} Y_{it}$ holds for all t $=1, \ldots, T_{\text{pre}}$. Then, we look for weights $\hat{\lambda}_t^{\text{sdid}}$ to equalize the time trend of the treatment sign and the time trend of the post-treatment. Finally, the average treatment effect $(\tau)$ is estimated under a two-dimensional fixed effects model:

$$\left( \hat{\tau}^{\text{sdid}}, \hat{\mu}, \hat{\alpha}, \hat{\beta} \right) = \underset{\tau, \mu, \alpha, \beta}{\operatorname{argmin}} \left\{ \sum_{i=1}^{N} \sum_{t=1}^{T} (Y_{it} - \mu - \alpha_i - \beta_t - W_{it}\tau)^2 \hat{\omega}_i^{\text{sdid}} \hat{\lambda}_t^{\text{sdid}} \right\}$$

*3.2. Designation of Variables*

3.2.1. Explained Variables

An integrated framework for cooperative innovation supports a cohesive community known as the innovation ecosystem. Within this system, each innovation entity engages in collaborative innovation with others, contributing to value co-creation by leveraging their unique roles. This collaboration is facilitated through the interaction and exchange of key innovation elements such as information, knowledge, technology, capital, and talent (Huang, 2023 [21]). To summarize, the key features of the innovation ecosystem are predominantly examined in the following dimensions: the Yangtze River Delta innovation ecosystem is able to gather a large number of main innovation bodies participating in regional innovation activities and supporting regional innovation functions within a certain range (Pekkarinen, 2006 [22]). These main innovation bodies include enterprises, higher education institutions, R&D institutions, governments, intermediary organizations, and other types of institutions. The innovation subjects in the Yangtze River Delta innovation ecosystem share common visions, goals, and interests; share knowledge and resources with each other; are interdependent; and create an "innovation community" through the establishment of bidirectional or multidirectional subject links and synergistic interactions.

A network of interactions is formed, linking innovation entities to their environment, through the circulation and sharing of materials, information, and energy (Zhang, 2018 [23]). The network structure strengthens ties among a variety of regional innovation entities, such as businesses, academic institutions, research organizations, and governmental bodies. It also ensures the efficient allocation and circulation of key innovation resources such as talent, capital, technology, and information. Consequently, the green technology innovation ecosystem indicators in this study are formulated with reference to the variable indicators from Song (2024) [24]. The entropy method is applied to quantify the specific values, as detailed in Table 1:

**Table 1.** Construction of green innovation ecosystem indicators.

| Level I Indicators | Secondary Indicators | Interpretation of Indicators | Direction of Indicators | Weighting of Secondary Indicators | Tier 1 Indicator Weights |
|---|---|---|---|---|---|
| Green technology innovation ecosystems | | | | | |
| | Pluralistic subjectivity | | | | 0.3895566 |
| | Number of enterprises | Size of regional green innovation firms | + | 0.042587 | |
| | Employees in the research and integrated technology services industry | Amount of regional research staff | + | 0.223399 | |
| | General public budget expenditure on science and technology | Strength of local government support for innovation activities | + | 0.123571 | |
| | Innovative resources | | | | 0.2825791 |
| | Number of persons enrolled in higher education | Amount of regional human capital | + | 0.105914 | |
| | Total credits for environmental projects | Financial investments in green innovation | + | 0.00997 | |
| | Number of patent applications for green inventions | Green innovation capacity | + | 0.166696 | |
| | Network infrastructure | | | | 0.0786705 |
| | Internet users | Level of network communication interconnection and interoperability among green innovation actors | + | 0.058537 | |
| | Total mileage of roads in country | Level of transportation access between various green innovation subjects | + | 0.020133 | |
| | Innovation environment | | | | 0.2491939 |
| | GDP per capita | Regional economic environment | + | 0.031917 | |
| | Number of occurrences of keywords "green" and "innovation" in government work reports | Regional policy environment | + | 0.031929 | |
| | Green coverage of built-up areas | Local ecology | + | 0.029829 | |
| | Total number of books in public libraries | Regional cultural environment | + | 0.002504 | |
| | Balance of deposits and loans of financial institutions | Regional financial environment | + | 0.059899 | |
| | Total value of foreign-invested enterprises | Regional open environment | + | 0.093117 | |

### 3.2.2. Explanatory Variables

To delve deeper into the effects of a watershed ecological compensation policy on the green technology innovation ecosystem in the area receiving compensation, this study introduces two dummy variables: one for time and another for city. One component is the time dummy variable (TIME). Using the watershed ecological compensation policy as the focus of evaluation, if the sample observations occur in the policy implementation year and later, time = 1; otherwise, time = 0. The second is the city dummy variable (treat). If a prefecture-level city is a watershed ecological compensation policy implementation area, treat = 1; otherwise, treat = 0. Based on this foundation, the interaction term (DID) serves as an explanatory variable to assess if city i experiences the effects of the watershed ecological compensation policy in period t. The interaction term (DID) is utilized to gauge the influence of the watershed ecological compensation policy on city i.

### 3.2.3. Control Variables

To avert endogeneity issues that could arise from neglecting variables that are relevant to the development of the green technology innovation ecosystem, a set of control variables is chosen, drawing on the research of scholars in the field: ① residents' saving capacity, assessed based on the equilibrium of residents' savings accounts (CNY 10,000); ② the current status of the industrial structure can be gauged by the proportion of the primary industry's value added to the GDP, which reflects the sector's contribution to the economy (%); ③ resource endowment, measured by the total amount of water resources (cubic meters); ④ urbanization level, assessed by the ratio of the non-agricultural population to the region's total population as of the year-end. (%); ⑤ population density, assessed by the ratio of the resident population to the urban planning area (%); and ⑥ sown area of crops, assessed by the year-end total of the region's arable land resources (thousand hectares). The detailed descriptions of various variables are shown in Table 2.

**Table 2.** Description of variables and composition of indicators.

|  | Variable Name | Explanation | Variable Symbol |
|---|---|---|---|
| explained variable | Green technology innovation ecosystems | Measurement of indices using the entropy method | Gtie |
| explanatory variable | Whether the watershed ecological compensation policy is implemented | Dummy variables (0,1) | Did |
| control variables | Savings capacity of the population | Balance of residential savings deposits | Rsa |
|  | Current status of industrial structure | Value added of primary sector as % of GDP | Csis |
|  | Resource endowment | Total water resources | Endo |
|  | urbanization level | Share of non-agricultural population in the total population of the region at the end of the year | Urle |
|  | population density | Ratio of resident population to urban zoning area | Pode |
|  | Sown crop area | Total resources of arable land at the end of the year | Paoc |
| intermediary variable | Carbon intensity | Carbon emissions by prefecture-level city | Cain |
|  | Sewage treatment rate | Efficiency of municipal wastewater treatment plants | Str |

### 3.3. Sources of Data and Descriptive Statistics

China's financial sector has grown rapidly due to its rudimentary development model, which has also led to escalating water pollution issues (Zhang, 2014 [25]). In this context, watershed pollution has been a difficult problem in China's water pollution management (Tang, 2022 [26]). Under the pilot agreement and the rollout of ecological compensation policies within the four major river basins, Huangshan City, Yulin City, Longyan City, and Ganzhou City have been identified as the principal regions that are eligible for compensation. Consequently, these four prefecture-level cities have been designated as the experimental group for the initiative (Song, 2023 [27]). Taking into account the impact of administrative boundaries on economic growth and recognizing that the river's headwaters

and midstream-area basins typically exhibit economic development characteristics that are akin to those of the treatment group, for this study, we chose 49 of the river's upstream and midstream sections' basins, located in provinces that are part of the experimental group and in adjacent provinces where river basin eco-compensation has not been implemented during the sampling period to serve as the control group. All relevant data for this analysis were sourced from the statistical records of the cities in question from 2005 to 2022.The data were obtained from the 2005–2022 City Statistics Bulletin and Yearbook, China Urban Construction Statistics Yearbook, and CSMAR database, and the patent information was sourced from the China Research Data Service Platform (CNRDS). To ensure the integrity of the data, linear interpolation was used to supplement a very small amount of missing data. The linear interpolation was based on constructing a straight line from two known data points, estimating the value of missing points using the coordinates and ratio of known points, and determining the position of missing points on the line segment by calculating the ratio. The linear interpolation method is suitable for situations where data change gently and is a simple and effective way to complete missing data. To mitigate the impact of absolute disparities among data points and the effects of outliers, and to alleviate the problem of heteroskedasticity between different variables, some variables were logarithmically or conjunctively processed to enhance the accuracy of the assessment results. Table 3 displays the statistical descriptions of the principal variables being examined.

**Table 3.** Descriptive statistical analysis.

| VARIABLE | (1)<br>N | (2)<br>Mean | (3)<br>sd | (4)<br>Min | (5)<br>Max |
|---|---|---|---|---|---|
| y | 954 | 0.0893 | 0.104 | 0.00948 | 0.710 |
| did | 954 | 0.0346 | 0.183 | 0 | 1 |
| Rsa | 954 | $1.287 \times 10^7$ | $1.513 \times 10^7$ | 636,497 | $1.251 \times 10^8$ |
| Csis | 954 | $3.266 \times 10^6$ | $8.888 \times 10^6$ | 62 | $1.039 \times 10^8$ |
| Endo | 954 | 14.82 | 7.642 | 1.220 | 38.08 |
| Urle | 954 | $1.034 \times 10^6$ | $1.072 \times 10^6$ | 51,475 | $6.273 \times 10^6$ |
| Pode | 954 | 312.1 | 213.6 | 24.95 | 994 |
| Paoc | 954 | 451.7 | 265.7 | 103 | 1440 |

## 4. Empirical Examinations

### 4.1. Benchmark Regression Results

In this research, the baseline estimation uses a two-way fixed effects model, and the specific regression results are presented in Table 4. Specifically, column (1) includes solely the primary explanatory variables; column (2) incorporates additional control variables; and column (3) encompasses all control variables. The three sets of regression analyses indicate that the dummy variable DID yields significantly positive regression coefficients at the 5% and 1% statistical significance levels. This result indicates that the implementation of watershed compensation policies markedly promotes the development of green technology innovation ecosystems in compensated cities. The regression analysis presented in column (3) reveals that the coefficient for the dummy variable DID is 0.0166. This implies that the adoption of the watershed compensation policy results in a 1.66% enhancement in the progression of green technology innovation ecosystems in cities receiving compensation, in contrast to those that do not. The data in Table 4 essentially confirm the primary research hypothesis that was put forth in this study, showing that the policy of watershed compensation does, indeed, play a role in bolstering the green technology innovation ecosystem. Hypothesis 1 has been validated.

**Table 4.** Benchmark regression results.

|  | (1) | (2) | (3) |
|---|---|---|---|
|  | y | y | y |
| did | 0.0276 *** | 0.0221 *** | 0.0166 ** |
|  | (3.49) | (6.97) | (2.35) |
| Rsa |  | $1.61 \times 10^{-9}$ *** | $1.55 \times 10^{-9}$ *** |
|  |  | (4.97) | (4.71) |
| Csis |  | $2.91 \times 10^{-9}$ *** | $2.90 \times 10^{-9}$ *** |
|  |  | (4.30) | (4.52) |
| Endo |  | 0.000374 | 0.000423 |
|  |  | (0.88) | (0.91) |
| Urle |  |  | $-2.95 \times 10^{-9}$ * |
|  |  |  | (−1.81) |
| Pode |  |  | −0.0000348 |
|  |  |  | (−1.65) |
| Paoc |  |  | 0.0000317 |
|  |  |  | (0.73) |
| _cons | 0.0321 *** | 0.0153 | 0.0142 |
|  | (9.46) | (1.44) | (0.64) |
| N | 795 | 795 | 795 |
| *individual fixed effect* | YES | YES | YES |
| *time-fixed effect* | YES | YES | YES |
| r2 | 0.587 | 0.805 | 0.807 |

Note: *, **, and *** indicate significance at the 10%, 5%, and 1% levels, respectively; same in the table below.

*4.2. Robustness Tests*

4.2.1. Parallel Trend Test

This research takes advantage of the "watershed compensation policy" as a quasi-natural experiment in green technological innovation ecosystems, and it uses the difference-in-differences method to evaluate the policy's impact on their development and expansion. In order to validate the robustness and credibility of the findings and to tackle possible endogeneity concerns like omitted variables and reverse causality, this study carries out a battery of tests. The main tests conducted include the counterfactual test and the parallel trend assessment.

The parallel trend test is essential for validating the difference-in-differences methodology, making sure that the treatment and control groups had comparable trends in climate change before the policy was enacted. Following the technique outlined in the research by Liu et al. (2024) [28], this paper performs the parallel trend test.

$$y_{it} = \alpha_0 + \sum_{k=-4}^{k=6} \omega_k DID_{i,t_0+k} + \xi X_{it} + \mu_i + \delta_t + \varepsilon_{it}$$

In this formula, k denotes the kth year following the introduction of the watershed compensation policy; $t_0$ represents the initiation year of the pilot watershed compensation policy in city i; $\xi$ denotes the parameter to be estimated for the control variables; and $DID_{i,t_0+k}$ signifies a set of dummy variables corresponding to the enactment of watershed compensation policies. Because the research sample design policy implementation occurred over a period of 18 years, involving more years than our study period, the data before the—fourth period and after the sixth period of the merger were used simultaneously to circumvent the issue of multicollinearity; the pilot policy before the first year was used as the base year, that is, in the regression model, to remove the dummy variables in the year. $\omega_k$ represents the key parameter, signifying the disparity in green technology innovation between the treatment and control groups in year k following policy implementation.

The parallel trend test presupposes that, without the watershed compensation policy, both the experimental and control groups would follow identical trends. Figure 2 displays the parallel trend chart, indicating that the regression coefficients before the rollout of

the watershed compensation policy were statistically insignificant. This suggests that prior to the policy's enactment, there was no substantial divergence in the developmental trajectories of the green technology innovation ecosystems between the treatment and control groups, exhibiting no systematic discrepancy. This result is consistent with the expectations of the parallel trend test. After the policy's implementation, the estimated coefficients turned significantly positive, showing a marked rise in values during the initial three periods post-implementation. This suggests that the watershed compensation policy has a compounding effect on the green technology innovation ecosystem, with its impact strengthening and intensifying over time. While the coefficients for the fourth and fifth periods remained positive, their values showed a declining trend, indicating a modest decline in the efficacy of the watershed compensation policy. However, the coefficients stabilized after the fifth period without any significant downward trend, indicating that the impact of the watershed compensation policy remained effective over the long term.

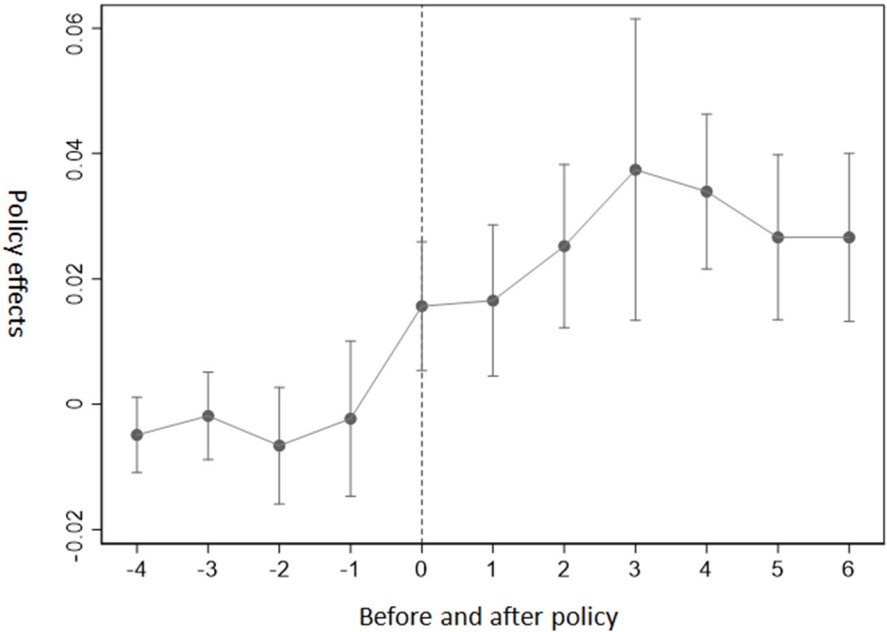

**Figure 2.** Parallel trend test.

4.2.2. Counterfactual Tests

To rule out the possibility that changes in fostering the growth of green technology innovation ecosystems in cities implementing watershed compensation policies are influenced by other simultaneous or recent macro-policies and, additionally, to substantiate the robustness of the findings, this study performs a counterfactual analysis by adjusting the dummy variables to reflect the policy's implementation timing. Supposing a consistent rollout of big data pilot zones worldwide, this study examines the estimated coefficients for the key explanatory variables. Placebo tests are conducted by advancing the pilot year of the comprehensive big data pilot zone by 2 years (column 1), 3 years (column 2), and 4 years (column 3), respectively, to examine whether the watershed compensation policy can still significantly influence the development of green technology innovation ecosystems in each compensated city. If the watershed compensation policy impact area is affected by other impacts, then the green technology innovation ecosystem in that place will not change significantly, regardless of whether the policy is implemented; then, the impact on the green technology innovation ecosystem of each test city should not change, regardless of advancing the policy implementation time by a few years. Conversely, it validates that the adoption of a watershed compensation policy is a significant determinant of green

technology innovation across the provinces under review. As shown in Table 5, the test outcomes reveal that the coefficients for the primary explanatory variable (DID) become non-significant when the timing of the watershed compensation policy's implementation is hypothetically set to an earlier stage. This demonstrates the robustness of the findings, confirming that the conclusions of this paper are not influenced by other factors and are therefore more reliable.

**Table 5.** Counterfactual tests.

| VARIABLE | (1)<br>2 Years in Advance | (2)<br>3 Years in Advance | (3)<br>4 Years in Advance |
|---|---|---|---|
| did | 0.007 | 0.002 | 0.000 |
| | (0.22) | (0.64) | (0.98) |
| Rsa | 0.000 *** | 0.000 *** | 0.000 *** |
| | (0.00) | (0.00) | (0.00) |
| Csis | 0.000 *** | 0.000 *** | 0.000 *** |
| | (0.00) | (0.00) | (0.00) |
| Endo | 0.000 | 0.000 | 0.000 |
| | (0.37) | (0.37) | (0.37) |
| Urle | −0.000 ** | −0.000 *** | −0.000 *** |
| | (0.01) | (0.01) | (0.00) |
| Pode | −0.000 | −0.000 | −0.000 |
| | (0.11) | (0.11) | (0.10) |
| Paoc | 0.000 | 0.000 | 0.000 |
| | (0.48) | (0.48) | (0.48) |
| Constant | 0.015 | 0.016 | 0.016 |
| | (0.50) | (0.49) | (0.49) |
| *individual fixed effect* | YES | YES | YES |
| *time-fixed effect* | YES | YES | YES |
| Observations | 795 | 795 | 795 |
| Number of id | 53 | 53 | 53 |
| R-squared | 0.807 | 0.807 | 0.807 |

Note: Columns (1), (2), and (3) display the regression outcomes of the pilot zone's establishment being advanced by 2, 3, and 4 years, respectively. *** $p < 0.01$, ** $p < 0.05$.

### 4.2.3. Placebo Test

Building upon the aforementioned test results, the model is simulated by randomly selecting samples of the experimental group for regression; specifically, a random selection of 60 cases from the overall sample constitutes the "pseudo-experimental group", instead of the real experimental group, for regression. The process is repeated 500 times, and 500 estimates are obtained accordingly. The probability density distribution of the estimated coefficients is displayed in Figure 3. The graph indicates that the estimated coefficients with random assignments are roughly around zero, following a normal distribution, and differ from the actual coefficient of 0.0166. This suggests that there is no substantial policy effect on the experimental group that was selected by chance. As a result, it is deduced that the influence of the watershed compensation policy on the green technology innovation ecosystems of the experimental group is authentic, and the shifts observed within these ecosystems can be genuinely ascribed to the policy's implementation. The absence of grave omitted variable issues in the model further attests to the robustness and dependability of the results obtained from the estimation.

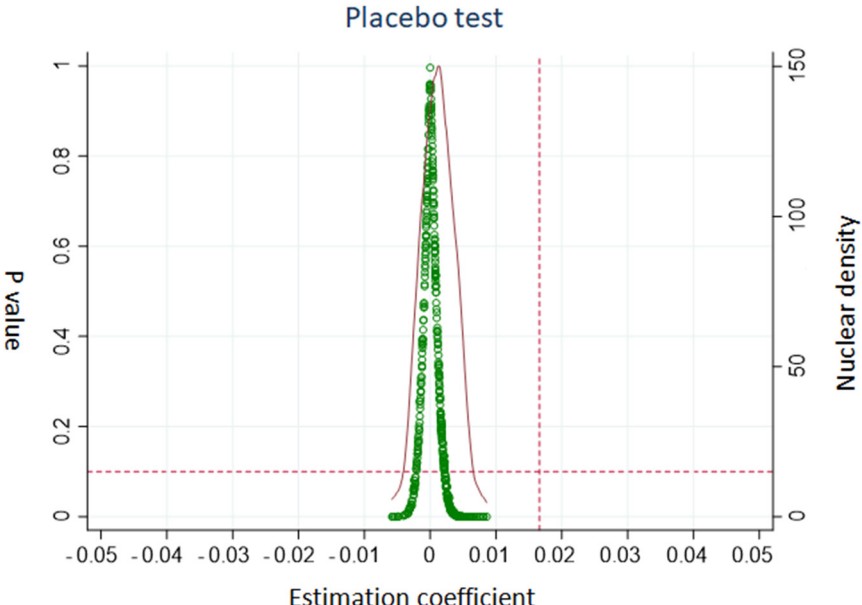

**Figure 3.** Individual placebo test.

### 4.2.4. Propensity Score Matching with Difference-in-Differences (DID) Analysis

To mitigate potential issues of endogeneity among variables and to assess the robustness of the empirical model, this research employs propensity score matching to address these challenges. Additionally, combining this method with a difference-in-differences approach minimizes sample disparities, enhancing the reliability of the model. In this research, cities that benefit from the watershed compensation policy are identified as the treatment group, while those that do not receive compensation constitute the control group. Using all control variables as covariates, propensity scores are estimated through the kernel matching method, matching the treatment group to the control group at a ratio of one to two. Figure 4 illustrates the standardized deviations of the covariates before and after matching. Examining the sample distribution before and after matching reveals significant differences in the sample distribution prior to matching, with a high degree of dispersion, while the standardized deviation of the samples after completing the kernel matching is significantly reduced, which are all reduced to less than 40%, and the variable selectivity bias can be alleviated. The results of the kernel density curves before and after matching are shown in Figure 5. Figure 5 demonstrates significant improvement in the alignment between the control and treatment groups after the matching process, which attests to the success of the matching strategy.

Building on the methodology of previous studies, we proceed to apply the double-difference model regression after conducting propensity score matching. Column (3) in Table 6 shows the regression results. The key explanatory variable, DID, maintains its significance at the 5% level, signifying that the policy of watershed compensation continues to have a notable impact on the sampled cities. These results align with the significance levels from the initial regression analysis, showing that the findings are robust post the PSM-DID assessment. This consistency further reinforces the reliability of the regression outcomes that are used as a reference.

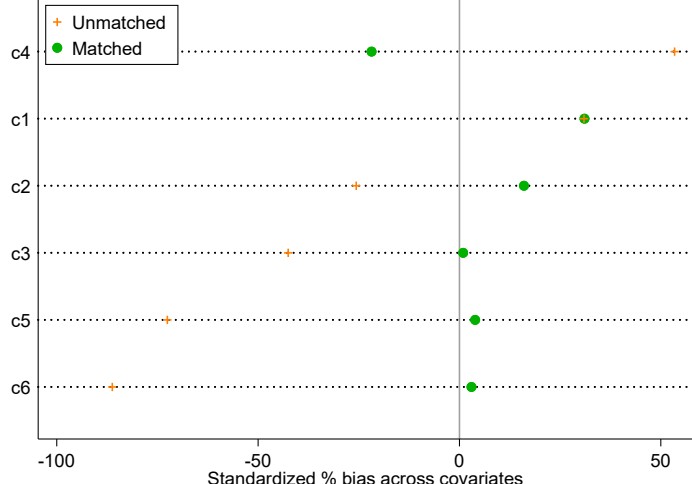

**Figure 4.** Covariate standardized deviation.

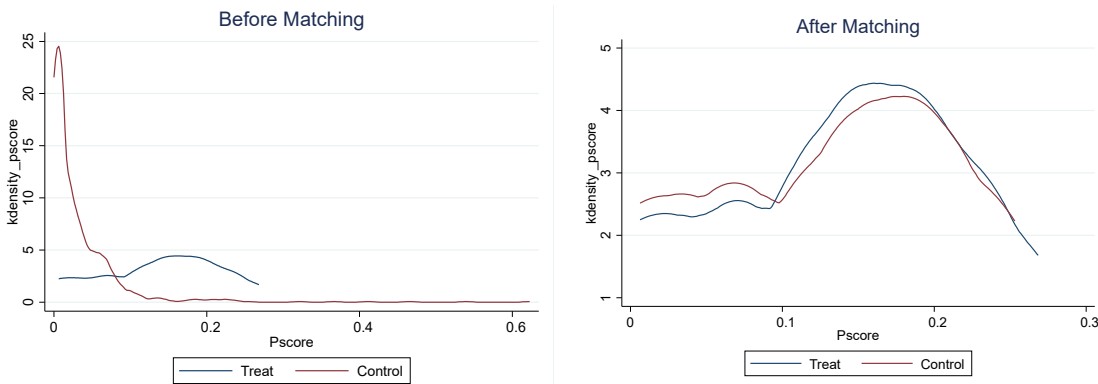

**Figure 5.** Propensity-matched kernel density curves.

**Table 6.** Other robustness tests.

| VARIABLE | (1) y | (2) y | (3) PSM y |
|---|---|---|---|
| did | 0.019 *** | 0.017 * | 0.007 ** |
| | (0.01) | (0.09) | (0.01) |
| Constant | 0.066 *** | 0.014 | 0.042 ** |
| | (0.00) | (0.22) | (0.01) |
| Control variables | YES | YES | YES |
| *Individual fixed effect* | YES | YES | YES |
| *Time-fixed effect* | YES | YES | YES |
| Observations | 930 | 954 | 631 |
| Number of id | 53 | 53 | 53 |
| R-squared | 0.869 | 0.807 | 0.795 |

pval in parentheses. *** $p < 0.01$, ** $p < 0.05$, * $p < 0.1$.

### 4.2.5. Other Robustness Tests

(1) Data filtering of the sample: To mitigate the impact of outliers on the study's conclusions, the core data were reanalyzed after trimming 5% of the extreme values. Column (1) in Table 7 displays the outcomes. After the elimination of outliers, the DID coefficient was still significantly positive at the 1% level, which was consistent with the baseline regression findings.

**Table 7.** Results of the mediation effect test.

| | (1)<br>y | (2)<br>Carbon Intensity | (3)<br>Sewage Treatment Rate |
|---|---|---|---|
| did | 0.0166 ** | −0.216 ** | 17.99 *** |
| | (2.35) | (0.01) | (3.22) |
| _cons | 0.0142 | 7.594 *** | 45.39 ** |
| | (0.64) | (0.00) | (2.48) |
| *N* | 954 | 954 | 754 |
| control variables | YES | YES | YES |
| *individual fixed effect* | YES | YES | YES |
| *time-fixed effect* | YES | YES | YES |
| r2 | 0.807 | 0.647 | 0.697 |

Robust pval in parentheses. *** $p < 0.01$, ** $p < 0.05$.

(2) Excluding contemporaneous policy interference: The opening of major transportation facilities such as high-speed rail stations will bring significant promotion effects in improving infrastructure construction, upgrading the green industrial structure and the introduction of talent cadres, etc. The development of green technology innovation ecosystems across various regions could be affected by the establishment of high-speed rail networks. During the implementation of the watershed compensation policy, several sample cities had high-speed rail stations established. To account for this, the study manually organizes data based on the high-speed rail network's opening timelines. Cities that launched high-speed rail in 2015, 2016, and 2017 are identified (with stations that opened after July 1 in any given year being attributed to the following year). These cities are then incorporated into the baseline regression by assigning policy dummy variables. For urban areas served by high-speed rail, the dummy variable is assigned a value of 1 from the year of service commencement and for all following years; for those without high-speed rail, the value is kept at 0. These policy dummy variables are included in the model to control the relevant policy's impact on the results. Table 6 shows the regression results. Compared with the benchmark regression, the core explanatory variable DID continues to exhibit a significantly positive coefficient, suggesting that other policies do not introduce bias into the estimation results, thereby reinforcing the conclusions of this study.

*4.3. Synthetic Control Method*

Figure 6 illustrates the variations in the green technology innovation ecosystem among the key cities benefiting from the watershed compensation policy and the corresponding synthetic control cities during the period of 2005–2022, with the dashed line representing the synthetic green technology innovation development level, the solid line representing the actual green technology innovation development level, and the vertical solid line indicating the timing of the policy intervention, marked as 2016 in this analysis. As can be seen from Figure 5, on the left side of the policy intervention time, the trajectory of actual green technological innovation development aligns with the growth trend of the synthetic green technological innovation development level, and the difference between the two groups is very small, which suggests that the synthetic control city closely aligns with the green technology innovation development level of the city receiving compensation before the introduction of the watershed policy. Following the policy's intervention period, on the right side, the two groups gradually deviate, and the actual green technological innovation ecosystem's development level is significantly higher than the synthetic green technological innovation ecosystem's development level, which implies that the hypothesis of this paper—i.e., that the policy of watershed compensation exerts a notably positive influence on the advancement of green technological innovation ecosystems, with a pronounced effect in the compensated cities—holds.

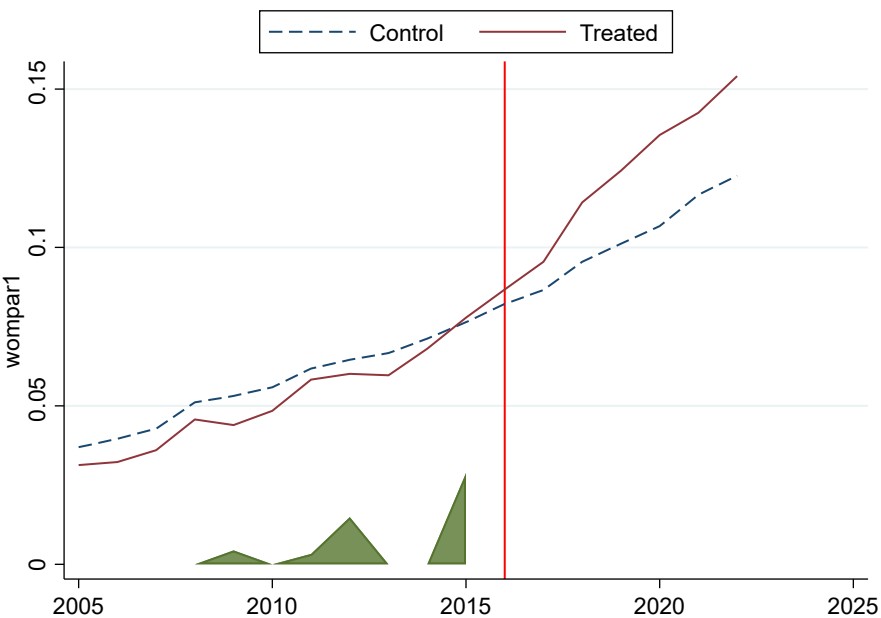

**Figure 6.** Comparison of actual and synthetic green technology innovation development paths.

*4.4. Mechanism Analysis*

4.4.1. Mediating Effects of Carbon Intensity

In order to verify the previous theoretical hypothesis that the carbon emission intensity can serve as an intermediary in the impact of the watershed compensation policy on the development of the green technology innovation ecosystem, this paper carries out an empirical test, the findings of which are presented in Table 8. As shown in column (2), the key explanatory variable DID shows a significantly negative coefficient at the 5% level. This suggests that the watershed compensation policy significantly contributes to the reduction in carbon emissions in cities that receive compensation. And emission reduction prompts enterprises to directly face the challenge of technological innovation. To achieve a reduction in carbon emissions, enterprises must develop and apply more efficient and cleaner production technologies (Li, 2021 [29]), and this process directly promotes the innovation of green technology. Consequently, the industry's overall energy structure improves, the energy intensity decreases, and $CO_2$ emissions are reduced. These actions not only mitigate the environmental impact but also bolster the construction of a green ecosystem. Lowering carbon emissions directly boosts the market demand for green technologies, as well. Taking advantage of this chance, businesses are encouraged to expedite the process of researching, developing, and marketing green technologies, aiming to secure a larger market share (Du, 2019 [30]); this, in turn, raises the level of innovation. To meet the emission reduction targets, the government often directly provides financial support or tax incentives to enterprises that are engaged in green technology innovation (Cai, 2020 [31]), which also directly promotes the cooperation between research institutions and enterprises, and this cooperative method swiftly enhances the process of green technology R&D and boosts the efficiency of innovation. In short, the decrease in carbon emissions directly affects the enterprise innovation motivation, market demand, policy incentives, and industry–university–research cooperation and other levels; consequently, there is a significant boost in the level of green technology innovation. Hypothesis H2a is confirmed.

**Table 8.** Heterogeneity regression results.

| VARIABLE | Low Public Expenditure (1) | High Public Expenditure (2) | Low Labor Productivity (3) | High Labor Productivity (4) | Resource-Based City (5) | Non-Resource-Based Cities (6) |
|---|---|---|---|---|---|---|
| did | 0.0151 | 0.0194 *** | −0.0236 | 0.0306 *** | 0.019 *** | −0.007 |
|  | (1.31) | (3.27) | (−0.79) | (7.50) | (4.26) | (−1.08) |
| _cons | 0.0324 *** | 0.0224 | 0.169 ** | 0.0266 *** | 0.064 *** | 0.025 |
|  | (2.78) | (0.79) | (2.42) | (3.22) | (7.99) | (1.11) |
| N | 375 | 378 | 378 | 375 | 342 | 420 |
| control variables | YES | YES | YES | YES | YES | YES |
| *individual fixed effect* | YES | YES | YES | YES | YES | YES |
| *time-fixed effect* | YES | YES | YES | YES | YES | YES |
| R-squared | 0.909 | 0.807 | 0.973 | 0.977 | 0.984 | 0.981 |

*** $p < 0.01$, ** $p < 0.05$.

### 4.4.2. Mediating Effects of Sewage Treatment Rates

Table 7, column (3), reveals that the key explanatory variable DID exhibits a coefficient that is significantly positive at the 1% significance level. This indicates that the watershed compensation policy greatly enhances the wastewater treatment efficiency in the compensated cities. Furthermore, this improvement in wastewater treatment efficiency has a substantial and multifaceted direct impact on the advancement of green technology innovation. Initially, the need to enhance the sewage treatment efficiency drives relevant companies to boost their R&D spending and commit to developing more efficient and energy-saving treatment technologies and equipment (Guo, 2019 [32]), quickening the development of green technology innovations; This demand-led momentum is significant. Secondly, the improvement in wastewater treatment efficiency is often accompanied by the discovery and application of new materials, such as biofilms and nanomaterials (Chong, 2010 [33]), and the research and development and use of these new materials directly enhance the level of innovation in green technologies. Furthermore, the improvement in wastewater treatment efficiency requires companies to continuously optimize and improve existing processes, and this continuous optimization process directly promotes technological innovation, making the treatment process more environmentally friendly and efficient (Lozano Avilés, 2019 [34]). In addition, the government's emphasis on wastewater treatment efficiency improvement is usually reflected through policy and financial support, and these direct support measures have incentivized enterprises to carry out green technological innovations (Pan, 2020 [35]), which has promoted technological progress. Additionally, the increase in wastewater treatment efficiency directly promotes the advancement of supporting technologies, such as environmental monitoring and data analytics. The innovation within these auxiliary technologies, in turn, contributes to raising the overall standard of green technology. The domain compensation policy has improved the efficiency of sewage treatment by enhancing the importance and technical structure of enterprise sewage treatment in pilot areas. Reducing the degree of water pollution has provided a more urgent background demand and cleaner experimental environment for green technology innovation, thereby stimulating relevant enterprises and research institutions to invest in research and development in sewage treatment and environmental protection technology and promoting the development of green technology innovation. Overall, the improvement in wastewater treatment efficiency directly plays a role in technology research and development, material innovation, process optimization, policy incentives, and the development of auxiliary technology and other aspects, which notably accelerates the enhancement of green technology innovation levels, thereby validating Hypothesis H2b.

### 4.5. Analysis of Heterogeneity Effects

#### 4.5.1. Heterogeneity at the Level of Public Financial Expenditure

Public financial expenditures refer to all kinds of payments that are made by the government using financial funds to satisfy societal public demands, to ensure national security, and to promote economic development and social progress (Wagner, 1958 [36]). These expenditures include government purchases of goods and services; transfer payments; and investments in public infrastructure, education, health, social security, and other areas. Public financial expenditure is an important means for the government to perform its functions, realize the optimal allocation of social resources through the rational allocation of financial resources, maintain social equity and justice, promote stable macroeconomic growth, and improve the standard of living of the nation (Curristine, 2007 [37]).

In this study, fiscal expenditure levels are reflected in the ratio of local fiscal spending to the Gross Domestic Product. The midpoint value of this metric serves as a cutoff, categorizing the cities into two segments: those with elevated fiscal outlays and those with reduced fiscal outlays. Distinct regression analyses are performed for each segment. As shown in Table 9, columns (1) and (2), it is evident that cities with greater public fiscal spending gain more from the watershed compensation policy, which markedly promotes the advancement of green technological innovation ecosystems within these municipalities. A possible reason is that high fiscal expenditure means that the government has a greater ability to provide funds to support green technology R&D, which includes providing subsidies, tax incentives, R&D grants, and other aspects of financial incentives (Bai, 2019 [38]), thus attracting more enterprises and research institutions to engage in green technology innovation activities. Government investment is not limited to direct financial support but also includes the construction and improvement of green infrastructure (Hannon, 2015 [39]), as well as investments in education and training systems, and these initiatives provide a solid foundation and the necessary talent pool for the promotion and utilization of green technologies. Meanwhile, government policy guidance and market demand creation measures, including the formulation of environmental standards and the advancement of green products, have further promoted the development of green technologies. In addition, the government shares innovation risks with enterprises, reduces enterprises' R&D costs through a risk-sharing mechanism (Liu, 2023 [40]), and improves the enthusiasm and success rate of enterprise innovation. The rise in public financial expenditure offers comprehensive support for green technology development. It not only fosters advancements in technological capabilities but also speeds up the adoption and widespread use of green technologies across various sectors of society. This, in turn, plays a crucial role in driving sustainable development. Hypothesis H3a is confirmed.

**Table 9.** Research hypotheses and testing results.

| Research Hypothesis | Can It Be Confirmed |
|---|---|
| **H1:** The policy of watershed compensation is posited to markedly boost the caliber of green technology innovation within compensated regions, thereby fortifying the evolution of the innovation ecosystem. | YES |
| **H2a**: Watershed offset policies can improve local innovation ecosystems by reducing carbon emissions in compensated areas. | YES |
| **H2b:** Watershed compensation policies can improve local innovation ecosystems by increasing wastewater treatment rates in compensated areas. | YES |
| **H3a:** Cities with greater public financial outlays experience a more pronounced impact of watershed compensation policies on their green technology innovation ecosystems. | YES |
| **H3b:** In cities where the labor productivity is higher, the impact of watershed compensation policies on green technology innovation ecosystems is more pronounced. | YES |
| **H3c:** In resource-based cities, watershed compensation policies have a more significant impact on the green technology innovation ecosystem. | YES |

4.5.2. Labor Productivity Heterogeneity

Labor productivity is the number of products or services that are produced per unit of labor input, and it is an important indicator of labor efficiency. It is typically measured by the quantity of goods that are manufactured or the economic value that is generated per hour of labor or per employee. A rise in labor productivity indicates that the same amount of labor input can produce a greater output, or that fewer labor hours are required to produce the same amount of products. It reflects the impact of technological progress, management optimization, labor quality improvement, and other factors in production efficiency and serves as a crucial metric for assessing the economic competitiveness and productivity of a nation or business.

As a result, the local labor productivity influences how watershed compensation policies affect the development of green technology innovation ecosystems in compensated cities. To analyze the heterogeneity caused by varying labor productivity levels, this study categorizes the labor productivity of each sample city into high and low groups based on the median value. Columns (3)–(4) of Table 8 display the results of the separate regressions that were performed for each group. The regression analysis shows that enacting watershed compensation policies significantly promotes the development of green technology innovation ecosystems in cities with elevated labor productivity. Conversely, this effect is not observed in the sample of cities with lower labor productivity. Typically, regions with elevated labor productivity also exhibit a higher degree of green technology adoption. This correlation arises because firms that are efficiency-driven are more inclined to integrate advanced technologies, encompassing green technologies, in their pursuit of optimizing resource use (Zhong, 2023 [41]). These enterprises not only have strong economic strength and profitability to bear the initial costs of green technology research and development, but also, their management level increases with productivity (Gumerov, 2020 [42]), which ensures the successful implementation and widespread adoption of green technology. Companies that boast high labor productivity recognize the significance of sustainable practices and consider environmental conservation to be crucial for their enduring business success. As a result, they are more likely to embrace green technologies that are aimed at mitigating environmental footprints (Chang, 2023 [43]). Located at the forefront of technology and management, they are able to quickly access and absorb the latest green technology information and tend to be the innovation leaders and green technology pioneers in their industries. Thus, the link between elevated labor productivity and advanced green technology development stems from a confluence of various elements. Hypothesis H3b has been validated.

4.5.3. Heterogeneity of Resource Endowment

On the basis of the above two heterogeneity analyses, this article also explores the heterogeneity of the impact of watershed compensation policies on green innovation systems from the perspective of the sample cities' resource endowments. Resource-based cities are assigned a value of 1, while non-resource-based cities are assigned a value of 0, and a regression analysis is conducted separately. According to columns (5)–(6) in Table 8, the core explanatory variable is significantly positive in the resource-based city sample but not significant in non-resource-based cities. In the sample of resource-based cities, the impact of watershed compensation policies on green technology innovation systems is stronger, while the effect is not significant in non-resource-based cities, which is mainly attributed to several key factors. Firstly, resource-based cities often have a high dependency on the extraction and processing of natural resources, which leads to relatively greater environmental pressures. The watershed compensation policy, through economic incentives, encourages these cities to pay more attention to environmental protection and innovation

in green technologies in the process of resource utilization. This policy orientation helps stimulate the innovation vitality of enterprises and increase research and development investments in environmental protection technology and production processes. Secondly, resource-based cities often face significant pressure to adjust their industrial structure, and watershed compensation policies provide these cities with opportunities for transformation. After the implementation of policies, enterprises will actively seek green technology innovation to reduce costs and improve production efficiency in order to cope with stricter environmental regulations. In contrast, non-resource-based cities rely less on natural resources and have a relatively diverse economic structure, which may not be as sensitive to watershed compensation policies as resource-based cities.

### 4.6. Summary of Empirical Results

After a series of empirical tests, many important conclusions have been drawn. The research hypotheses and testing results are shown in Table 9.

## 5. Conclusions and Policy Suggestions

Utilizing panel data from major cities across four major river basins spanning the years of 2005 to 2022, this research conducts an empirical analysis to assess the impact of watershed compensation policies on green technology innovation ecosystems within selected urban areas. The analysis is conducted with the aid of double-difference models, synthetic double-difference models, and spatial double-difference models. The main conclusions of the article include the following: (1) The development of green technology innovation ecosystems in cities receiving compensation is notably enhanced by the watershed compensation policy, a finding that holds up after undergoing several robustness checks and being evaluated with synthetic control double-difference methodologies. (2) The analysis of the mechanisms reveals that the substantial decrease in carbon emission intensity and the improvement in sewage treatment rates are key pathways through which the watershed compensation policy enhances the green technology innovation ecosystem. It is recommended to broaden the development path of the green technology innovation ecosystem by concentrating on these two aspects. (3) The progression of the ecosystem for green technology innovation based on the watershed compensation policy is characterized by significant heterogeneity, and the effect is more significant in cities with high public financial expenditure and high labor productivity. Cities with high public financial expenditure aid in refining the allocation pattern of public fiscal resources so that the financial pressure on education and green technology research is lower, which helps to invest in R&D capital and create a good policy environment. In areas characterized by high labor productivity, an equivalent amount of capital investment yields higher returns, thereby boosting the efficiency of corporate R&D and the adoption of green innovation technologies. This, in turn, fosters a more conducive research environment for the advancement of green technology innovation ecosystems.

Drawing from these insights, this study offers the following recommendations:

Initially, refine the watershed compensation policies. Strengthen the establishment and continuous backing of ecological compensation measures for river basins and augment the allocation of funds dedicated to ecological compensation, explore market-oriented compensation methods, and encourage the use of non-government funds to set up ecological compensation funds and other financial methods to improve the ecological compensation standard in order to stimulate the large pilot areas to carry out ecological environmental protection initiatives and build enthusiasm. The parallel trend test outcomes indicate that the watershed compensation policy exerts its effectiveness in the later phase, but there is a clear trend of an insufficient effect, so it is necessary to make targeted improvements to the implementation process according to the actual situation and sustain the application of watershed compensation to amplify its impact. Enhancing the connectivity and collabora-

tion between the upper and lower reaches of the basin in areas such as capital, technology, human resources, and policy is essential. Additionally, it is crucial to develop a variety of compensation methods to sustain the ongoing application of ecological policies within the basin's pilot regions.

Secondly, it is essential to vigorously enhance the combined efforts of pollution reduction and emission reduction. Key to the advancement of a green technology innovation ecosystem is the decrease in carbon emissions and an improvement in the efficiency of wastewater treatment. For one thing, a carbon market should be established, and emissions reduction should be promoted through carbon emissions trading. Develop green transportation, promote electric vehicles and public transportation, enhance forests' carbon sink capacity, and formulate strict carbon emission regulations. In addition, it is imperative to refine the energy structure by advocating for renewable energy sources, diminishing the reliance on fossil fuels, and enhancing energy efficiency. Additionally, the promotion of energy-saving and emission-reducing technologies and products is crucial, as is the cultivation of a market demand for green technologies. In terms of sewage treatment, the key lies in technology upgrades and adopting energy-efficient treatment technologies, such as MBR. Implement wastewater resourcing, recover resources from sludge, optimize treatment systems, and improve the operational efficiency. Strengthen monitoring and control to ensure that the water quality meets discharge standards and gain policy support and public participation. Efficient and broad green product markets and clear public goals can propel the power and confidence of green technological innovation of enterprises, thus creating a good R&D environment and social environment and improving the construction of a green technological innovation ecosystem.

Thirdly, it is important to address regional disparities judiciously. Tailored to the specific developmental context of each region, we should establish appropriate and varied ecological compensation standards and initiatives. Maximizing the efficacy of the funds that are designated for the green technology innovation ecosystem should be a priority. Due to the lower levels of financial expenditure and lower labor productivities of some cities, to achieve a reasonable formulation of the rules, policymakers should avoid the "one size fits all" approach during the policy formulation process, and it is essential to consider the environmental resource disparities and other regional factors to craft tailored development strategies. This approach aims to leverage regional strengths and optimize the exploitation of local resources to their fullest potential.

However, improving the green technology innovation system through watershed compensation policies may face a series of complex and multidimensional challenges in the future. The high cost, insufficient technological maturity, and limited public awareness and acceptance, as well as the complexity of policy formulation and implementation, are all urgent issues that need to be addressed. The research and development, production, and initial application of green technology products are usually accompanied by high costs, which to some extent limit their market penetration. Meanwhile, some green technologies are still in the early stages of development, and their maturity and stability need to be further improved. In addition, the level of public awareness of the importance of green technology is directly related to its broad application prospects. At the policy level, how to ensure that watershed compensation policies can accurately and effectively stimulate green technology innovation, while properly handling conflicts of interest among all parties and avoiding regional policy discomfort, will be a crucial issue. Therefore, in the future, it is necessary to comprehensively consider the above factors and take more comprehensive and systematic measures to effectively promote the continuous improvement and development of the green technology innovation system. Finally, this article only discusses the four major river basins in China. However, these results may not be applicable to other regions with different ecological, economic, and institutional

backgrounds. Therefore, further exploration of this issue will continue to address the policy needs of regions with different developmental backgrounds.

**Author Contributions:** Conceptualization, M.L. and J.Z.; methodology, J.Z.; software, J.Z.; validation, H.D., M.L. and J.Z.; formal analysis, J.Z.; investigation, M.L.; resources, M.L.; data curation, M.L.; writing—original draft preparation, M.L.; writing—review and editing, J.Z.; visualization, J.Z.; supervision, J.Z.; project administration, J.Z.; funding acquisition, J.Z. All authors have read and agreed to the published version of the manuscript.

**Funding:** This research was funded by Guangxi Philosophy and Social Science Self-financed Project 'Research on the Path and Mechanism of China-ASEAN Digital Trade Co-operation Driving the Development of Digital Economy in Guangxi' (2023FGL025).

**Data Availability Statement:** Data can be obtained from the corresponding author.

**Conflicts of Interest:** The authors declare no conflict of interest.

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
