# Peer review of "A Study on the Impact of Watershed Compensation Policies on Green Technology Innovation Ecosystems"

_systems, doi:10.3390/systems13010044_

Round 1
Reviewer 1 Report
Comments and Suggestions for Authors
This paper discusses the impact of basin compensation policy on green technology innovation ecosystem, and uses the staggered differential model and the synthetic control differential model for analysis. However, the following problems remain:
1. Introduction
The introduction does not make clear the correlation between basin ecological compensation and green innovation ecosystem, that is, it does not make clear the urgency and innovation of the research problem.
2. Theoretical analysis and hypothesis
The theoretical mechanism is relatively weak, and the impact of basin compensation policy on green technology innovation ecosystem cannot be deduced. The relevant theories should be reinterpreted, the logical deduction process of the hypothesis should be refined, and the theoretical explanation of the rationality of the hypothesis should be added.
3. Research methods
This paper uses the staggered differential model for research, and the synthetic control differential model is only used as a robustness test, so it is not necessary to list.
4. Empirical results and analysis
There are many problems in the empirical process, mainly as follows:
(1) It is unacceptable that the empirical form is in Chinese.
(2) The formula in line 338 refers to the first 3 periods and the last 6 periods of the policy, but the parallel trend test is the first 4 periods and the last 6 periods of the policy, which is contradictory.
(3) Some forms can be merged.
(4) The influence mechanism lacks the analysis of mediator variables.
(5) The heterogeneity analysis lacks typical heterogeneity variables, such as different regions, and the heterogeneity variables conducted by the author are not typical.
5. Conclusions and policy recommendations
Make policy suggestions concrete and put forward more operational measures. At the same time, discussions on possible challenges and countermeasures in the course of policy implementation can be increased.
6. Language and format
Refine some statements to avoid verbosity and repetition. At the same time, check the full text for grammar and spelling errors to ensure the rigor of the article.
7. It is also unacceptable that the paper lacks innovative description and there are no innovative points in the whole paper.
Comments on the Quality of English LanguageRefine some statements to avoid verbosity and repetition. At the same time, check the full text for grammar and spelling errors to ensure the rigor of the article.
Author Response
Response to Reviewer 1 Comments
Dear Editor and Reviewers:
Thank you for your letter and for the reviewers’ comments concerning our manuscript entitled “Study on the Impact of Watershed Compensation Policies on Green Technology Innovation Ecosystems” (systems-3367084). Those comments are all valuable and very helpful for revising and improving our paper, as well as the important guiding significance to our researches. We have studied comments carefully and have made correction which we hope meet with approval. Revised portion are marked in red in the manuscript. The main corrections in the paper and the responds to the reviewer’s comments are as flowing:
Responds to the reviewer’s comments:
Comment 1: The introduction does not make clear the correlation between basin ecological compensation and green innovation ecosystem, that is, it does not make clear the urgency and innovation of the research problem.
Response 1: Thank you for your suggestion. We have added a paragraph in the introduction section on the correlation between watershed compensation policies and green innovation ecosystems.
The supplementary content is as follows: Nations in economic transition prioritize innovation as a key to harmonizing ecological and economic interests, a crucial and evident role. Addressing the pronounced imbalance, establishing a green tech innovation ecosystem is essential for advancing sustainable economic growth. This ecosystem, under strategic innovation guidance, exhibits enhanced coordination, self-regulation, and openness, becoming a central force in industry reorganization and economic shift. Amid economic challenges, understanding the impact of watershed compensation policies on green innovation is vital. These policies, as economic incentives, balance regional ecological and economic interests, spurring corporate green tech innovation. Amidst escalating climate and environmental crises, balancing watershed conservation with economic greening is an urgent theoretical and practical challenge. The synergy between watershed policies and green innovation ecosystems offers a fresh approach to this issue, showcasing the potential for policy and tech integration. This alliance fortifies watershed conservation and supports the growth of green innovation ecosystems, laying a foundation for economic transformation and sustainability.
Comment 2: The theoretical mechanism is relatively weak, and the impact of basin compensation policy on green technology innovation ecosystem cannot be deduced. The relevant theories should be reinterpreted, the logical deduction process of the hypothesis should be refined, and the theoretical explanation of the rationality of the hypothesis should be added.
Resopnse2: Thank you for your suggestion. In order to increase the rationality of the theoretical explanation of the hypothesis, we have improved the theoretical mechanism part and perfected the logical derivation of the research hypothesis. The supplementary content is as follows:
- Theoretical Analysis and Hypothesis Development
2.1. Influence of Watershed Compensation Policies on the Green Innovation Ecosystems in Compensated Regions
Companies serve as pivotal catalysts for regional technological advancements and green innovation solutions. Research by Shen Neng (2012) 13 and others indicates that enterprises' profit and production functions are intricately linked to their emission levels and environmental protection expenditures. Specifically, pollution emissions positively correlate with output revenue and negatively with environmental protection expenditures, suggesting that increased investment in environmental protection leads to reduced emissions.The implementation of eco-compensation initiatives, particularly watershed compensation policies, significantly impacts green innovation systems. These policies not only incentivize enterprises to invest in pollution control, mitigating environmental degradation and fostering the "technological progress effect of pollution control," but also encourage technological innovation through financial incentives. The "innovation compensation effect" arises as upstream enterprises are rewarded for adopting environmentally friendly practices that benefit downstream areas, motivating them to innovate not solely for compliance but also to capitalize on government incentives.
Moreover, the integration of watershed compensation policies into green innovation systems promotes collaboration among upstream and downstream enterprises, as well as between enterprises and government agencies. This collaborative framework facilitates the sharing of best practices, the development of novel technologies, and the optimization of resource use, collectively contributing to a more sustainable and innovative production process. Based on these premises, the inaugural hypothesis of this study is put forth:
H1: The policy of watershed compensation is posited to markedly boost the caliber of green technology innovation within the compensated regions, thereby fortifying the evolution of the innovation ecosystem.
2.2. Mechanisms of Impact of Watershed Compensation Policies on Innovation Ecosystems in Compensated Areas
Under the cross-provincial horizontal ecological protection compensation mechanism, local governments reevaluate the equilibrium between economic growth and ecological preservation, thereby strengthening environmental regulations and implementing measures for both end-of-pipe treatment and source prevention to reduce water pollutant emissions in accordance with agreed-upon watershed water quality standards. Notably, adopting a source prevention and control approach for water pollution treatment can synergistically drive CO2 reduction. Within the framework of inter-provincial river basin horizontal ecological protection compensation systems, local governments vigorously promote cleaner production (Wang, 2018[8]). This, in turn, encourages enterprises to utilize cleaner energy, enhance energy conservation, and augment investments in green innovations. Consequently, the industry's overall energy structure improves, energy intensity decreases, and CO2 emissions are reduced. These actions not only mitigate the environmental impact but also bolster the construction of a green ecosystem. Illustratively, the Xin'an River Basin has optimized and upgraded over 510 projects during the first two rounds of the agreement, investing 6 billion yuan in constructing a circular economy park with integrated heating, desalination, and sewage treatment systems. This has facilitated the establishment of a green industry system, resulting in significant economic growth—with GDP surpassing 50 billion- and 60-billion-yuan milestones—and financial income exceeding 200 million yuan. This exemplifies the qualitative and effective transformation of green resources into economic benefits, achieving substantial improvements in ecological, economic, and social outcomes.
Similarly, the Jiuzhou River Basin has upgraded nine water-related industrial enterprises as of 2022, implemented "one factory, one case" pollution control for 16 water-related enterprises, and constructed a 73.33-hectare small and medium-sized enterprise industrial transfer park in the Upper Jiuzhou River basin. By guiding water-related enterprises to relocate from the riverbanks into the park and supporting the development of green environmental protection industries, the basin has promoted industrial transformation and upgrading. These initiatives have contributed to reducing carbon emissions, thereby enhancing the resilience and sustainability of the green ecosystem(Chen, 202214).
Conversely, within the framework of ecological compensation for inter-provincial river basins on a horizontal level, local governments can restrict the development of enterprises in water pollution-intensive industries, which are often high carbon-intensive as well (Wang, 20188). This regulation not only controls the market entry of related enterprises but also promotes the development of the entire industry's energy structure and intensity towards CO2 reduction. Additionally, enhancing wastewater treatment rates through compensation mechanisms further refines green ecosystem construction. For instance, the Xin'an River Basin enforces an industrial entry negative list system and invests heavily in pollution prevention, resulting in the closure and relocation of numerous polluting enterprises. Similarly, the Jiuzhou River Basin shuts down polluting industries and refuses to approve wastewater-discharging projects (Tu 201215). Moreover, the Hanjiang-Tingjiang River Basin and Longyan City in Fujian Province conduct extensive remediation projects, which include improving wastewater treatment facilities to enhance treatment rates. By integrating wastewater treatment improvements into ecological compensation strategies, these regions effectively bolster their green ecosystems, ensuring sustainable development and environmental protection.The second hypothesis of this paper can be proposed:
H2a: Watershed offset policies can improve local innovation ecosystems by reducing carbon emissions in compensated areas.
H2b: Watershed compensation policies can improve local innovation ecosystems by increasing wastewater treatment rates in compensated areas.
2.3. Diversity in the Influence of Watershed Compensation Policies on the Development of Green Technology Innovation Ecosystems
In regions with high public fiscal expenditures and high labor productivity, the effectiveness of river basin compensation in enhancing green ecosystem construction is more pronounced. High fiscal revenues enable governments to allocate more funds towards long-term development projects (Yan, 202416), such as education and scientific research, which directly support green technology R&D and innovation. This fosters a conducive environment for technological advancements, providing sustained funding and intellectual resources. Furthermore, high labor productivity not only augments firm profits but also encourages investments in technological innovation (Brynjolfsson, 201117), as firms recognize the correlation between productivity gains and enhanced competitiveness. With a skilled and efficient workforce, enterprises possess the human capital necessary for technological breakthroughs, further supporting green ecosystem development (Fedulova, 201918). Thus, the combination of robust public fiscal expenditures and high labor productivity synergistically amplifies the impact of river basin compensation on green ecosystem construction. The advantages of resource-based cities in terms of the impact of watershed compensation policies on green technology innovation ecosystems are mainly reflected in the fact that they can obtain financial support through compensation policies for green technology innovation; As a provider and protector of ecological services, there is a stronger willingness to improve resource utilization efficiency and achieve sustainable development through technological innovation. Based on this, the following hypothesis is proposed:
H3a: Cities with greater public financial outlays experience a more pronounced impact of watershed compensation policies on their green technology innovation ecosystems.
H3b: In cities where labor productivity is higher, the impact of watershed compensation policies on green technology innovation ecosystems is more pronounced.
H3c: In resource-based cities, watershed compensation policies have a more significant impact on the green technology innovation ecosystem.
Following the above ideas, this article constructs a mechanism framework for the impact of watershed compensation policies on the green technology innovation ecosystem, as shown in Figure 1.
Figure 1. Mechanism diagram
Comment 3: This paper uses the staggered differential model for research, and the synthetic control differential model is only used as a robustness test, so it is not necessary to list.
Resopnse3: Thank you for your suggestion. The manuscript uses a comprehensive control DID model to re synthesize the experimental group samples to test the effectiveness of the policy. Therefore, a comprehensive control DID method was added on the basis of interweaving DID, making the article more rigorous and scientific.
Comment 4: Empirical results and analysis
(1) It is unacceptable that the empirical form is in Chinese. The formula in line 338 refers to the first 3 periods and the last 6 periods of the policy, but the parallel trend test is the first 4 periods and the last 6 periods of the policy, which is contradictory. (2) Some forms can be merged. (3) The influence mechanism lacks the analysis of mediator variables.
Resopnse4(1): Thank you for the reviewer's suggestions and for identifying our typographical errors. We have verified the corresponding number and made the necessary modifications.
Resopnse4(2): Thank you for your suggestion. As per the reviewer's request, we have merged the PSM-DID test results into Table 6 in the manuscript, with the following modifications:
Table 6. Other robustness tests.
|
|
(1) |
(2) |
(3)PSM |
|
VARIABLES |
y |
y |
y |
|
|
|
|
|
|
did |
0.019*** |
0.017* |
0.007** |
|
|
(0.01) |
(0.09) |
(0.01) |
|
Constant |
0.066*** |
0.014 |
0.042** |
|
|
(0.00) |
(0.22) |
(0.01) |
|
control variables |
YES |
YES |
YES |
|
individual fixed effect |
YES |
YES |
YES |
|
time fixed effect |
YES |
YES |
YES |
|
Observations |
930 |
954 |
631 |
|
Number of id |
53 |
53 |
53 |
|
R-squared |
0.869 |
0.807 |
0.795 |
pval in parentheses. *** p<0.01, ** p<0.05, * p<0.1.
Resopnse4(3): Thank you for your feedback. We have added an analysis of mediator variables in the mechanism analysis to make it more comprehensive. The improved mechanism analysis is as follows:
4.4.1. Mediating Effects of Carbon Intensity
In order to verify the previous theoretical hypothesis that carbon emission intensity can serve as an intermediary in the impact of the watershed compensation policy on the development of the green technology innovation ecosystem., this paper carries out an empirical test, the findings are presented in Table 8. As shown in Column (2), the key explanatory variable DID show a significantly negative coefficient at the 5% level. This suggests that the watershed compensation policy significantly contributes to the reduction of carbon emissions in cities that receive compensation. And emission re-duction prompts enterprises to directly face the challenge of technological innovation, to achieve a reduction in carbon emissions, enterprises must develop and apply more efficient and cleaner production technology (Li, 2021[28]), this process directly promotes the innovation of green technology. Consequently, the industry's overall energy structure improves, energy intensity decreases, and CO2 emissions are reduced. These actions not only mitigate the environmental impact but also bolster the construction of a green ecosystem. Lowering carbon emissions directly boosts the market demand for green technologies, secondly. Taking advantage of this chance, businesses are encour-aged to expedite the process of researching, developing, and marketing green tech-nologies, aiming to secure a larger market share. (Du, 2019[29]), this, in turn, raises the level of innovation. To meet the emission reduction targets, the government often di-rectly provides financial support or tax incentives to enterprises engaged in green technology innovation (Cai, 2020[30]), which also directly promotes the cooperation between research institutions and enterprises, and this cooperative method swiftly enhances the process of green technology R&D and boosts the efficiency of innovation. In short, the decrease in carbon emissions directly affects the enterprise innovation motivation, market demand, policy incentives and industry-university-research cooperation and other levels, consequently, there is a significant boost in the level of green technology innovation. Hypothesis H2a is confirmed.
4.4.2. Mediating Effects of Sewage Treatment Rates
Table 7, column (3), reveals that the key explanatory variable DID exhibits a co-efficient that is significantly positive at the 1% significance level. This indicates that the watershed compensation policy greatly enhances wastewater treatment efficiency in the compensated cities. Furthermore, this improvement in wastewater treatment efficiency has a substantial and multifaceted direct impact on the advancement of green technology innovation. Initially, the need to enhance sewage treatment efficiency drives relevant companies to boost their R&D spending and commit to developing more efficient and energy-saving treatment technologies and equipment (Guo, 2019[31]), quickening the development of green technology innovations, this demand-led mo-mentum is significant. Secondly, the improvement of wastewater treatment efficiency is often accompanied by the discovery and application of new materials, such as biofilms and nanomaterials (Chong, 2010[32]), and the research and development and use of these new materials directly enhance the level of innovation of green technologies. Furthermore, the improvement of wastewater treatment efficiency requires companies to continuously optimize and improve existing processes, and this continuous optimiza-tion process directly promotes technological innovation, making the treatment process more environmentally friendly and efficient (Lozano Avilés, 2019[33]). In addition, the government's emphasis on wastewater treatment efficiency improvement is usually reflected through policy and financial support, and these direct support measures have incentivized enterprises to carry out green technological innovations (Pan, 2020[34]), which has promoted technological progress. Additionally, the increase in wastewater treatment efficiency directly promotes the advancement of supporting technologies, such as environmental monitoring and data analytics. The innovation within these auxiliary technologies, in turn, contributes to raising the overall standard of green technology. The domain compensation policy has improved the efficiency of sewage treatment by enhancing the importance and technical structure of enterprise sewage treatment in pilot areas. Reducing the degree of water pollution has provided a more urgent demand background and cleaner experimental environment for green technology innovation, thereby stimulating relevant enterprises and research institutions to invest in research and development in sewage treatment and environmental protection technology, and promoting the development of green technology innovation. Overall, the improvement of wastewater treatment efficiency directly plays a role in technology research and development, material innovation, process optimization, policy incentives and the development of auxiliary technology and other aspects, which notably accelerates the enhancement of green technology innovation levels, thereby validating Hypothesis H2b.
Comment 5: The heterogeneity analysis lacks typical heterogeneity variables, such as different regions, and the heterogeneity variables conducted by the author are not typical.
Resopnse5: Thank you for your suggestion. Based on the ideas you provided, we have added resource endowment heterogeneity analysis in the heterogeneity analysis section, which divides the sample cities into resource-based cities and non-resource-based cities for regression analysis to verify the differences in the effectiveness of watershed compensation policies for different resource endowment cities. The analysis section is as follows:
4.5.3. Heterogeneity of resource endowment
On the basis of the above two heterogeneity analyses, this article also explores the heterogeneity of the impact of watershed compensation policies on green innovation systems from the perspective of sample city resource endowments. Resource based cities are assigned a value of 1, while non-resource-based cities are assigned a value of 0, and regression analysis is conducted separately. According to columns (5) - (6) in Table 8, the core explanatory variable is significantly positive in the resource-based city sample, but not significant in non-resource-based cities. In the sample of resource-based cities, the impact of watershed compensation policies on green technology innovation systems is stronger, while the effect is not significant in non-resource-based cities, mainly attributed to several key factors. Firstly, resource-based cities often have a high dependence on the extraction and processing of natural resources, which leads to relatively greater environmental pressure. The watershed compensation policy, through economic incentives, encourages these cities to pay more attention to environmental protection and innovation of green technologies in the process of resource utilization. This policy orientation helps to stimulate the innovation vitality of enterprises and in-crease research and development investment in environmental protection technology and production processes. Secondly, resource-based cities often face significant pres-sure to adjust their industrial structure, and watershed compensation policies provide these cities with opportunities for transformation. After the implementation of policies, enterprises will actively seek green technology innovation to reduce costs and improve production efficiency, in order to cope with stricter environmental regulations. In con-trast, non-resource-based cities rely less on natural resources and have a relatively di-verse economic structure, which may not be as sensitive to watershed compensation policies as resource-based cities.
Comment 6: Make policy suggestions concrete and put forward more operational measures. At the same time, discussions on possible challenges and countermeasures in the course of policy implementation can be increased.
Resopnse6:
Thank you for your valuable suggestion. We have improved the policies and recommendations, and we have added the following content:
However, improving the green technology innovation system through watershed compensation policies may face a series of complex and multidimensional challenges in the future. The high cost, insufficient technological maturity, limited public awareness and acceptance, as well as the complexity of policy formulation and implementation, are all urgent issues that need to be addressed. The research and development, pro-duction, and initial application of green technology products are usually accompanied by high costs, which to some extent limits their market penetration. Meanwhile, some green technologies are still in the early stages of development, and their maturity and stability need to be further improved. In addition, the level of public awareness of the importance of green technology is directly related to its broad application prospects. At the policy level, how to ensure that watershed compensation policies can accurately and effectively stimulate green technology innovation, while properly handling conflicts of interest among all parties and avoiding regional policy discomfort will be a crucial issue. Therefore, in the future, it is necessary to comprehensively consider the above factors and take more comprehensive and systematic measures to effectively promote the continuous improvement and development of the green technology in-novation system.
Comment 7: Refine some statements to avoid verbosity and repetition. At the same time, check the full text for grammar and spelling errors to ensure the rigor of the article.
Resopnse7: Thank you for your suggestion. We checked the grammar and spelling errors throughout the manuscript to ensure its rigor.
Comment 8: It is also unacceptable that the paper lacks innovative description and there are no innovative points in the whole paper.
Resopnse8: Thank you for your suggestion. We have added a description of the innovative points of the article at the end of the introduction. The added paragraph is as follows:
Based on the pilot policy of "watershed compensation", a quasi natural experiment was established. Compared with existing literature, the marginal contribution of this article lies in: firstly, through the mechanism analysis of the watershed compensation policy on the green technology innovation ecosystem, it was found that the watershed compensation policy improves the construction level of the green technology innovation ecosystem in the pilot area by reducing carbon emissions and improving sewage treatment efficiency; The second is to explore the significant heterogeneity in the impact of watershed compensation policies on green innovation systems at the levels of public expenditure and labor productivity. Thirdly, the article has a certain degree of innovation in the selection of explanatory and mediating variables. And the innovative combination of synthetic control method and interleaved dual activation model validates the robustness of the research content in the article.
We appreciate for Editor/Reviewers’ warm work earnestly, and hope that the correction will meet with approval.
Once again, thank you very much for your comments and suggestions.

Reviewer 2 Report
Comments and Suggestions for Authors
This paper studies the impact of watershed compensation policies on the green technology innovation ecosystem by using methods such as difference-difference method. On the whole, the article is more standardized, but there are still some problems, mainly concentrated in the following points.
1. In line 324, in addition to the symbol for significance, please also indicate the meaning of the numbers in brackets in the table.
2. The lower summation limit of the parallel trend test equation constructed in line 338 is set to -3. However, in the picture on line 366, the longest year before the policy was implemented is -4. To maintain the consistency and accuracy of the data, adjust the lower summation limits in the equation to ensure that they match the range of data shown in the chart.
3. The contents of the table on line 543 should be expressed in English and not in any other language.
4. The literature on the topic of ecological compensation cited in lines 726 and 727 cannot be found in a database similar to Web of Science (WOS). To ensure the authority and accuracy of your citations, please refer to articles from reputable journals. For example, the following article topic and ecological compensation, subordinate to the journal's more authority: https://doi.org/10.1016/j.envres.2023.118074
Author Response
Response to Reviewer 2 Comments
Dear Editor and Reviewers:
Thank you for your letter and for the reviewers’ comments concerning our manuscript entitled “Study on the Impact of Watershed Compensation Policies on Green Technology Innovation Ecosystems” (systems-3367084). Those comments are all valuable and very helpful for revising and improving our paper, as well as the important guiding significance to our researches. We have studied comments carefully and have made correction which we hope meet with approval. Revised portion are marked in red in the manuscript. The main corrections in the paper and the responds to the reviewer’s comments are as flowing:
Responds to the reviewer’s comments:
Comment 1: In line 324, in addition to the symbol for significance, please also indicate the meaning of the numbers in brackets in the table.
Response 1: Thank you for your suggestion. We have added explanations in the comments section below the table, where the number in parentheses represents the Z value.
Comment 2: The lower summation limit of the parallel trend test equation constructed in line 338 is set to -3. However, in the picture on line 366, the longest year before the policy was implemented is -4. To maintain the consistency and accuracy of the data, adjust the lower summation limits in the equation to ensure that they match the range of data shown in the chart.
Response 2: Thank you very much for your valuable comments. In order to maintain the consistency and accuracy of the data, we adjust the sum of the lower limits in the equation in the manuscript to ensure that they match the data range shown in the chart.
Comment 3: The contents of the table on line 543 should be expressed in English and not in any other language.
Response 3: Thank you for your suggestion. This was an error in our translation. Once again, thank you for your discovery.
Comment 4: The literature on the topic of ecological compensation cited in lines 726 and 727 cannot be found in a database similar to Web of Science (WOS).
Response 4: Thank you for your suggestion. We have added articles from more authoritative journals as references in the manuscript, such as: https://doi.org/10.1016/j.envres.2023.118074
We appreciate for Editor/Reviewers’ warm work earnestly, and hope that the correction will meet with approval.
Once again, thank you very much for your comments and suggestions.

Reviewer 3 Report
Comments and Suggestions for Authors
The paper addresses an important and timely issue—how watershed compensation policies influence green technology innovation ecosystems—which is highly relevant for sustainability and environmental economics. The study fills a gap in the literature regarding the link between ecological compensation policies and green innovation.
The use of staggered Difference-in-Differences (DID) and Synthetic DID models enhances the paper's methodological rigor. The authors conduct comprehensive robustness checks such as placebo tests, counterfactual analyses, and propensity score matching (PSM-DID), strengthening the reliability of the findings.
The paper effectively identifies two key pathways—carbon intensity reduction and wastewater treatment efficiency—to explain how compensation policies impact innovation ecosystems.
The analysis considers heterogeneity based on public financial expenditure and labor productivity, showing that policy effects are not uniform across regions. This provides policy-relevant insights for targeted interventions.
The paper concludes with practical and specific recommendations to improve watershed policies and foster green innovation, which is valuable for policymakers and stakeholders.
We enumerate below some weaknesses of the manuscript. The interpolation method used for missing data is not explained in sufficient detail.
While the paper conducts robustness tests to mitigate endogeneity, the discussion on causality remains somewhat limited.
The study focuses on four major river basins in China. However, the results might not be generalizable to other regions with different ecological, economic, and institutional contexts. This limitation is not discussed in the paper.
While the authors identify carbon emission reduction and wastewater treatment as mechanisms, additional mechanisms such as industrial restructuring or clean energy adoption could be explored further. Some sections, particularly the methodology and robustness tests, are highly technical and dense. Clearer explanations, possibly with simplified tables or flowcharts, would make the paper more accessible to a broader audience.
The empirical results section could benefit from a summary table highlighting the main findings for each hypothesis.
The research methodology does not have references.
Author Response
Response to Reviewer 3 Comments
Dear Editor and Reviewers:
Thank you for your letter and for the reviewers’ comments concerning our manuscript entitled “Study on the Impact of Watershed Compensation Policies on Green Technology Innovation Ecosystems” (systems-3367084) . Those comments are all valuable and very helpful for revising and improving our paper, as well as the important guiding significance to our researches. We have studied comments carefully and have made correction which we hope meet with approval. Revised portion are marked in red in the manuscript. The main corrections in the paper and the responds to the reviewer’s comments are as flowing:
Responds to the reviewer’s comments:
Comment 1: The interpolation method used for missing data is not explained in sufficient detail.
Response 1: Thank you for your suggestion. We have added a description of linear interpolation in the data source section to make it more complete. The supplementary content is as follows:
To ensure the integrity of the data, linear interpolation is used to supplement a very small amount of missing data. Linear interpolation is based on constructing a straight line from two known data points, estimating the value of missing points using the coordinates and ratio of known points, and determining the position of missing points on the line segment by calculating the ratio. The linear interpolation method is suitable for situations where data changes gently and is a simple and effective way to complete missing data. To mitigate the impact of absolute disparities among data points and the effects of outliers, and to alleviate the problem of heteroskedasticity between different variables, some variables are logarithmically or conjunctively processed to enhance the accuracy of the assessment results.
Comment 2: While the paper conducts robustness tests to mitigate endogeneity, the discussion on causality remains somewhat limited.
Response 2: Thank you for your suggestion. According to the reviewer's comments, the discussion of causal relationships has been improved in the manuscript and added to the mechanism analysis to strengthen causal analysis. The content is as follows:
4.4. Mechanism Analysis
4.4.1. Mediating Effects of Carbon Intensity
In order to verify the previous theoretical hypothesis that carbon emission intensity can serve as an intermediary in the impact of the watershed compensation policy on the development of the green technology innovation ecosystem., this paper carries out an empirical test, the findings are presented in Table 8. As shown in Column (2), the key explanatory variable DID show a significantly negative coefficient at the 5% level. This suggests that the watershed compensation policy significantly contributes to the reduction of carbon emissions in cities that receive compensation. And emission re-duction prompts enterprises to directly face the challenge of technological innovation, to achieve a reduction in carbon emissions, enterprises must develop and apply more efficient and cleaner production technology (Li, 2021[28]), this process directly promotes the innovation of green technology. Consequently, the industry's overall energy structure improves, energy intensity decreases, and CO2 emissions are reduced. These actions not only mitigate the environmental impact but also bolster the construction of a green ecosystem. Lowering carbon emissions directly boosts the market demand for green technologies, secondly. Taking advantage of this chance, businesses are encouraged to expedite the process of researching, developing, and marketing green technologies, aiming to secure a larger market share. (Du, 2019[29]), this, in turn, raises the level of innovation. To meet the emission reduction targets, the government often directly provides financial support or tax incentives to enterprises engaged in green technology innovation (Cai, 2020[30]), which also directly promotes the cooperation between research institutions and enterprises, and this cooperative method swiftly enhances the process of green technology R&D and boosts the efficiency of innovation. In short, the decrease in carbon emissions directly affects the enterprise innovation motivation, market demand, policy incentives and industry-university-research cooperation and other levels, consequently, there is a significant boost in the level of green technology innovation. Hypothesis H2a is confirmed.
4.4.2. Mediating Effects of Sewage Treatment Rates
Table 7, column (3), reveals that the key explanatory variable DID exhibits a co-efficient that is significantly positive at the 1% significance level. This indicates that the watershed compensation policy greatly enhances wastewater treatment efficiency in the compensated cities. Furthermore, this improvement in wastewater treatment efficiency has a substantial and multifaceted direct impact on the advancement of green technology innovation. Initially, the need to enhance sewage treatment efficiency drives relevant companies to boost their R&D spending and commit to developing more efficient and energy-saving treatment technologies and equipment (Guo, 2019[31]), quickening the development of green technology innovations, this demand-led momentum is significant. Secondly, the improvement of wastewater treatment efficiency is often accompanied by the discovery and application of new materials, such as biofilms and nanomaterials (Chong, 2010[32]), and the research and development and use of these new materials directly enhance the level of innovation of green technologies. Furthermore, the improvement of wastewater treatment efficiency requires companies to continuously optimize and improve existing processes, and this continuous optimization process directly promotes technological innovation, making the treatment process more environmentally friendly and efficient (Lozano Avilés, 2019[33]). In addition, the government's emphasis on wastewater treatment efficiency improvement is usually reflected through policy and financial support, and these direct support measures have incentivized enterprises to carry out green technological innovations (Pan, 2020[34]), which has promoted technological progress. Additionally, the increase in wastewater treatment efficiency directly promotes the advancement of supporting technologies, such as environmental monitoring and data analytics. The innovation within these auxiliary technologies, in turn, contributes to raising the overall standard of green technology. The domain compensation policy has improved the efficiency of sewage treatment by enhancing the importance and technical structure of enterprise sewage treatment in pilot areas. Reducing the degree of water pollution has provided a more urgent demand background and cleaner experimental environment for green technology innovation, thereby stimulating relevant enterprises and research institutions to invest in research and development in sewage treatment and environmental protection technology, and promoting the development of green technology innovation. Overall, the improvement of wastewater treatment efficiency directly plays a role in technology research and development, material innovation, process optimization, policy incentives and the development of auxiliary technology and other aspects, which notably accelerates the enhancement of green technology innovation levels, thereby validating Hypothesis H2b.
Comment 3: The study focuses on four major river basins in China. However, the results might not be generalizable to other regions with different ecological, economic, and institutional contexts. This limitation is not discussed in the paper.
Response 3: Thank you for your suggestion. We have already supplemented the shortcomings of the paper at the end, and further research is needed on regions with different ecological, economic, and institutional backgrounds. This is also the direction of our future research. The supplementary content is as follows:
However, improving the green technology innovation system through watershed compensation policies may face a series of complex and multidimensional challenges in the future. The high cost, insufficient technological maturity, limited public awareness and acceptance, as well as the complexity of policy formulation and implementation, are all urgent issues that need to be addressed. The research and development, pro-duction, and initial application of green technology products are usually accompanied by high costs, which to some extent limits their market penetration. Meanwhile, some green technologies are still in the early stages of development, and their maturity and stability need to be further improved. In addition, the level of public awareness of the importance of green technology is directly related to its broad application prospects. At the policy level, how to ensure that watershed compensation policies can accurately and effectively stimulate green technology innovation, while properly handling conflicts of interest among all parties and avoiding regional policy discomfort will be a crucial issue. Therefore, in the future, it is necessary to comprehensively consider the above factors and take more comprehensive and systematic measures to effectively promote the continuous improvement and development of the green technology in-novation system. Finally, this article only discusses the four major river basins in China. However, these results may not be applicable to other regions with different ecological, economic, and institutional backgrounds. Therefore, further exploration of this issue will continue to address the policy needs of regions with different development back-grounds.
Comment 4: While the authors identify carbon emission reduction and wastewater treatment as mechanisms, additional mechanisms such as industrial restructuring or clean energy adoption could be explored further. Some sections, particularly the methodology and robustness tests, are highly technical and dense. Clearer explanations, possibly with simplified tables or flowcharts, would make the paper more accessible to a broader audience.
Response 4: Thank you for your suggestion. Following your suggestion, in order to increase readers' understanding of this article, we have added a mechanism diagram to enable a wide range of readers to comprehend the paper. The supplementary content is as follows:
Figure 1. Mechanism diagram
Comment 5: The empirical results section could benefit from a summary table highlightin5g the main findings for each hypothesis.
Response 5: Thank you for your suggestion. Based on your suggestion, we have added a table summarizing the empirical results at the end of the empirical analysis to demonstrate whether the research hypothesis has been confirmed. The supplementary content is as follows:
4.6. Summary of Empirical Results
After a series of empirical tests, many important conclusions have been drawn. The research hypotheses and testing results are shown in the table below.
Table 9. Research hypothesis and testing situation.
|
research hypothesis |
Can it be confirmed |
|
H1: The policy of watershed compensation is posited to markedly boost the caliber of green technology innovation within the compensated regions, thereby fortifying the evolution of the innovation ecosystem. |
YES |
|
H2a: Watershed offset policies can improve local innovation ecosystems by reducing carbon emissions in compensated areas. |
YES |
|
H2b: Watershed compensation policies can improve local innovation ecosystems by increasing wastewater treatment rates in compensated areas. |
YES |
|
H3a: Cities with greater public financial outlays experience a more pronounced impact of wa-tershed compensation policies on their green technology innovation ecosystems. |
YES |
|
H3b: In cities where labor productivity is higher, the impact of watershed compensation poli-cies on green technology innovation ecosystems is more pronounced. |
YES |
|
H3c: In resource-based cities, watershed compensation policies have a more significant impact on the green technology innovation ecosystem. |
YES |
Comment 6: The research methodology does not have references.
Response 6: Thank you for your suggestion. We have added references in the research methodology section.
We appreciate for Editor/Reviewers’ warm work earnestly, and hope that the correction will meet with approval.
Once again, thank you very much for your comments and suggestions.

Round 2
Reviewer 1 Report
Comments and Suggestions for Authors
The paper is well revised and ready for publication
Comments on the Quality of English LanguageNo more comments on English writing
Reviewer 3 Report
Comments and Suggestions for Authors
The authors responded to the reviewers' comments.